# Inhibiting Endothelial Cell Function in Normal and Tumor Angiogenesis Using BMP Type I Receptor Macrocyclic Kinase Inhibitors

**DOI:** 10.3390/cancers13122951

**Published:** 2021-06-12

**Authors:** Jin Ma, Jiang Ren, Midory Thorikay, Maarten van Dinther, Gonzalo Sanchez-Duffhues, Josselin Caradec, Pascal Benderitter, Jan Hoflack, Peter ten Dijke

**Affiliations:** 1Department of Cell and Chemical Biology, Leiden University Medical Center, 2333 ZC Leiden, The Netherlands; J.Ma@lumc.nl (J.M.); ren-jiang@outlook.com (J.R.); M.Thorikay@lumc.nl (M.T.); M.A.H.van_Dinther@lumc.nl (M.v.D.); G.Sanchez_Duffhues@lumc.nl (G.S.-D.); 2Oncode Institute, Leiden University Medical Center, 2333 ZC Leiden, The Netherlands; 3Oncodesign S.A., 21000 Dijon, France; JCARADEC@oncodesign.com (J.C.); PBENDERITTER@oncodesign.com (P.B.); JHOFLACK@oncodesign.com (J.H.)

**Keywords:** activin receptor-like kinase, angiogenesis, bone morphogenetic protein, endothelial cell, macrocyclic kinase inhibitor

## Abstract

**Simple Summary:**

Anti-angiogenesis agents have shown anti-cancer activity by preventing blood vessel ingrowth, thereby limiting tumour growth and metastasis. Although these molecules lead to prolonged overall survival of cancer patients, therapy resistance is easily acquired. Therefore, novel inhibitors against other signaling pathways mediating angiogenesis are needed to achieve more efficient and sustainable targeting of the angiogenesis process. Here, we synthesized and identified two compounds belonging to a new class of small molecules termed macrocyclics that selectively inhibit bone morphogenetic protein receptor kinase activity. One compound also inhibits vascular endothelial growth factor-induced signalling. Treatment studies using in vitro cultured cells and zebrafish embryos revealed that both compounds impaired endothelial cell function and decreased normal and tumour-induced angiogenesis. Both compounds might provide a steppingstone for the development of novel-angiogenesis therapeutic agents.

**Abstract:**

Angiogenesis, i.e., the formation of new blood vessels from pre-existing endothelial cell (EC)-lined vessels, is critical for tissue development and also contributes to neovascularization-related diseases, such as cancer. Vascular endothelial growth factor (VEGF) and bone morphogenetic proteins (BMPs) are among many secreted cytokines that regulate EC function. While several pharmacological anti-angiogenic agents have reached the clinic, further improvement is needed to increase clinical efficacy and to overcome acquired therapy resistance. More insights into the functional consequences of targeting specific pathways that modulate blood vessel formation may lead to new therapeutic approaches. Here, we synthesized and identified two macrocyclic small molecular compounds termed OD16 and OD29 that inhibit BMP type I receptor (BMPRI)-induced SMAD1/5 phosphorylation and downstream gene expression in ECs. Of note, OD16 and OD29 demonstrated higher specificity against BMPRI activin receptor-like kinase 1/2 (ALK1/2) than the commonly used small molecule BMPRI kinase inhibitor LDN-193189. OD29, but not OD16, also potently inhibited VEGF-induced extracellular regulated kinase MAP kinase phosphorylation in ECs. In vitro, OD16 and OD29 exerted strong inhibition of BMP9 and VEGF-induced ECs migration, invasion and cord formation. Using Tg (*fli*:EGFP) zebrafish embryos, we found that OD16 and OD29 potently antagonized dorsal longitudinal anastomotic vessel (DLAV), intra segmental vessel (ISV), and subintestinal vessel (SIV) formation during embryonic development. Moreover, the MDA-MB-231 breast cancer cell-induced tumor angiogenesis in zebrafish embryos was significantly decreased by OD16 and OD29. Both macrocyclic compounds might provide a steppingstone for the development of novel anti-angiogenesis therapeutic agents.

## 1. Introduction

Angiogenesis is an intricately regulated multistep process in which new blood vessels form from pre-existing vessels. It is critical during embryonic development and later to maintain the homeostasis of almost all normal tissues. Moreover, it plays a pivotal role in the development of some chronic diseases, for example, neoplasia and cardiovascular disorders [1,2,3]. It has been demonstrated that angiogenesis is mainly guided by endothelial cells (ECs), the one-cell thick continuous tubular structure forming the inner surface of blood vessels [4]. The EC behavior is regulated by several signaling pathways, and the most well-studied one is the vascular endothelial cell growth factor (VEGF) pathway [5,6]. Secreted VEGF binds to its tyrosine kinase VEGF receptor 2 (VEGFR2) to activate downstream signaling pathways, including mitogen-activated protein kinase (MAPK)/extracellular-signal-regulated kinase (ERK), phosphoinositide 3-kinases (PI3 Ks), and AKT to mediate biological responses, such as stimulation of EC proliferation, migration, sprouting and network formation [7,8]. Besides VEGF, numerous angiogenic regulators are identified, such as fibroblast growth factor (FGF), platelet-derived growth factor (PDGF), and transforming growth factor-β (TGF-β) family members [9]. The exerted effects of these factors are highly dependent on particular EC types, their differentiation state, and the cellular context [9,10].

Bone morphogenetic proteins (BMPs), which belong to the TGF-β family, have a critical function in regulating vascular development during embryonic angiogenesis, post-embryonic wound healing, and also in promoting vesiculation-related diseases development [11]. Of the BMP family, BMP2, BMP4, and BMP6 promote ECs proliferation, migration, and sprouting [4,12,13]. Interestingly, BMP9 was found to increase proliferation and tube formation in human umbilical vein endothelial cells (HUVECs), while it has an opposite function on some other EC types, such as human dermal microvascular endothelial cells (HMVEC-d) [14,15].

Like other TGF-β family members, BMPs exert their cellular effects by inducing heteromeric complexes of selective type II and type I serine/threonine kinase receptors. The type I receptor is phosphorylated by type II kinase receptor and acts downstream of the type II receptor. Upon activation of BMP type I receptors (BMPRIs), intracellular signaling is triggered by the phosphorylation of SMAD1/5/8, which together with SMAD4 translocate into the nucleus to regulate the transcription of downstream genes, such as ID1 and SMAD6, [14,16,17]. Four BMPRI have been described: activin receptor-like kinase 1 (ALK1), ALK2, ALK3, and ALK6. BMP2 and BMP4 bind with high affinity to ALK3 and ALK6, while BMP6 and BMP7 mainly signal via ALK2 [11]. It was shown that the signaling activated via ALK2/3/6 through the expression of constitutively active forms of ALK2 (caALK2), caALK3, or caALK6 in bovine aortic endothelial cells (BAECs) promotes EC migration and tube formation [18]. BMP9 has a high affinity for ALK1, which is primarily expressed in activated ECs [19]. BMP9-ALK1 mediated signaling is critical for vascular system development [19,20,21]. Similar to BMP9, the role of ALK1 mediated signaling in angiogenesis is highly context-dependent and incompletely understood. Inactivation of ALK1 in mice leads to embryonic lethality because of severe vascular abnormalities, indicating the crucial function of ALK1 in controlling angiogenesis [22]. Seki et al. created a null mutant mouse line for *ALK1* (*ALK1* ^lacZ^) to track the dynamic expression changes of ALK1 during embryonic development and physiological processes as well as angiogenesis in tumor growth [23]. The relatively high expression of ALK1 in embryonic arterial endothelium and newly formed arterial vessels in tumors indicates the important role of ALK1 in arterialization [23]. In contrast, activated ALK1 signaling inhibits retinal vascularization by interacting with the NOTCH pathway [24].

Multiple agents targeting angiogenesis have been shown to inhibit pro-angiogenic pathways or stimulate anti-angiogenetic responses, and some have been approved by the United States (US) Food and Drug Administration (FDA) for therapy of cancer patients. Most anti-angiogenic agents are designed to target VEGF signaling [25]. The treatment with anti-angiogenic agents not only blocks the formation of new vessels to limit blood supplies to tumor cells but also stabilizes mature tumor blood vessels. The latter is referred to as vessel normalization [26,27,28]. These normalized tumor vessels may prevent the transition of the tumor to a more aggressive, invasive phenotype induced by a hypoxic environment and also facilitate the delivery of chemotherapeutics to the tumor due to reduced vessel leakiness [29]. Frequently, anti-angiogenic drugs are given together with chemo- or targeted therapy. However, treatment effects are often short-lived due to the acquired anti-angiogenic drug resistance [30]. Thus, the development of inhibitors targeting other signaling pathways required for angiogenesis is urgently needed to overcome drug resistance. BMP signaling inhibitors, including antibodies and small molecules targeting tumor angiogenesis, are in development but have not reached FDA approval yet [31,32].

A problem for small molecule inhibitor development is the lower specificity to targets compared to neutralizing antibodies. Off-target effects may limit their application in the clinic. For example, the best characterized BMPRI inhibitor LDN-193189 inhibits not only ALK1/2 but also other BMPRIs and even other kinases [33]. An innovative way to improve the selectivity and specificity of small molecules involves the macrocyclization of small molecule compounds. In essence, the principle is simple: the macrocyclization of kinase hinge-binding fragments is used to generate potent and highly selective small molecules with low molecular weight and attractive physicochemical properties. The cyclic linker induces the rigid, distinct three-dimensional shape of the inhibitor that is complementary in shape to the ATP-binding site in kinases [34].

In this study, we synthesized and identified two macrocyclic molecules OD16 and OD29, that have much higher specificity in blocking ALK1 and ALK2 as compared to LDN-193189, which targets all the TGF-β family type I receptors. Interestingly, OD29 also has inhibitory effects on VEGF signaling, indicating that it targets two pathways involved in angiogenesis. OD16 and OD29 demonstrated potent anti-angiogenic activity both in vitro, as measured by EC assays, and in vivo, as measured the vessel formation during zebrafish embryonic development and tumor angiogenesis triggered by breast cancer cells in zebrafish. The newly developed BMPRI inhibitors provide new agents for interrogating the function of BMPIs and may lead to the development of anti-angiogenesis treatment for potential clinical application.

## 2. Materials and Methods

### 2.1. Cell Culture

Human EA.hy926 endothelial cells (which were was established by fusing primary human umbilical vein cells with a thioguanine-resistant clone of human A549 adenocarcinoma cells) were routinely cultured with Dulbecco’s modified Eagle’s medium (DMEM, 11965092, ThermoFisher Scientific, Waltham, MA, USA) supplemented with 10% fetal bovine serum (FBS, 16000044, ThermoFisher Scientific), and 100 IU/mL penicillin/streptomycin. Human umbilical vein endothelial cells (HUVECs) were cultured in Medium 199 (M199, 31150022, ThermoFisher Scientific) supplemented with 20% FBS, 100 IU/mL penicillin/streptomycin, 0.01 IU/mL Heparine (DF2451, LEO pharma), and 94.2 μg/mL Bovine Pituitary Extract (BPE, 13028014, ThermoFisher Scientific). All the endothelial cells were cultured on 1% (*w/v*) gelatin (G1890, Sigma-Aldrich, St. Louis, MI, USA) pre-coated flask/plates and maintained in a 5% CO_2_ humidified air incubator at 37 °C. The cells were regularly tested for the absence of mycoplasma infection.

### 2.2. Reagents

BMP9 (3209-BP/CF) was purchased from R&D (Minneapolis, MN, USA). Recombinant BMP6, and TGF-β2 were a kind gift from Slobodan Vukicevic (University of Zagreb, Zagreb, Croatia) and Joachim Nickel (University of Wurzburg, Würzburg, Germany), respectively. LDN-193189 was purchased from Selleckchem (Houston, Texas, USA). VEGFR kinase inhibitors Axitinib (PZ0193) and Sunitinib (PZ0012) were purchased from Sigma-Aldrich (Saint Louis, MI, USA).

### 2.3. Luciferase Reporter Assay

The BMP responsive element (BRE)-driven transcriptional luciferase reporter, which contains SMAD1/5-SMAD4 responsive elements of the *ID1* gene promoter, was used to assess the activation of the BMP/SMAD signaling pathway [35]. A lentiviral vector carrying this reporter construct (pRRL-BRE-Luc) was generated by digesting the MLP-BRE-Luc plasmid with *Kpn*I and *Sal*I and then inserted into the lentivirus vector pRRL-CMV digested with *Xho*I and *Kpn*I. Lentiviral particles were made using HEK 293 T cells, and EA.hy926 cells were transduced to generate BRE-Luc stably expressing cells, named as EA.hy926-BRE-Luc.

EA.hy926-BRE-Luc cells were seeded at approximately 5 × 10^5^ cells/well in a 24-well plate. Cells were serum-starved overnight by incubating with a serum-free medium. After starvation, the cells were pre-incubated with the compounds for 30 min and subsequently stimulated overnight with ligands (or vehicle control). Thereafter, the cells were lysed, and luciferase activity was measured with the firefly luciferase assay System kit (E1501, Promega Corporation, Madison, WI, USA) using a PerkinElmer luminometer. The firefly luciferase activity was normalized to protein content, which was determined using the DC protein assay kit (5000111, Bio-Rad, Hercules, CA, USA).

### 2.4. Western Blot Analysis

EA.hy926 cells or HUVECs were seeded in 6-well plates and starved in a serum-free medium for 4 h after reaching 80% confluence. The compounds were added 30 min prior to stimulation with the corresponding ligands. EA.hy926 cells were incubated with BMP9 (1 ng/mL), BMP6 (50 ng/mL), TGF-β2 (1 ng/mL), or vehicle control for 1 h. For HUVECs, after 4 h starvation, the cells were pre-treated with or without the indicated compounds for 30 min and subsequently stimulated with VEGF (40 ng/mL) for 10 min. The cells were harvested and lysed in radioimmunoprecipitation assay (RIPA) lysis buffer containing 1× complete protease inhibitor cocktail (11836153001, Roche, Basel, Switzerland). The protein concentration was determined using the DC protein assay kit (5000111, Bio-Rad). After adding loading buffer, the samples were boiled and proteins separated by sodium dodecyl sulfate polyacrylamide gel electrophoresis (SDS-PAGE). Proteins in the gel were transferred to polyvinylidene difluoride (PVDF) membranes (IPVH00010, Merck Millipore, Burlington, MA, USA) and immunoblotted with 1000 times diluted primary antibodies: phospho-SMAD1/5 (9516 S, Cell Signaling Technology, Leiden, The Netherlands), phospho-SMAD2 (pSMAD2, home-made) [36], SMAD1 (6944 S, Cell Signaling Technology), SMAD2 (3103 S, Cell Signaling Technology), phospho-extracellular signal-regulated kinase/mitogen-activated protein kinase (pERK/MAPK, 4370, Cell Signaling Technology), mitogen-activated protein kinase/ERK1/2 (ERK1/2/MAPK, 9102 S, Cell Signaling Techonol), VEGFR (2479, Cell Signaling), pVEGFR (2478, Cell Signaling Technology), α/β-Tubulin (2148, Cell Signaling Technology), glyceraldehyde 3-phosphate dehydrogenase (GAPDH, MAB374, Merck Millipore). Clarity™ enhanced chemiluminescence (ECL) reagent (1705060, Bio-Rad) was used to visualize secondary antibodies conjugated to horseradish peroxidase (HRP). GAPDH or Tubulin was used as protein loading controls. All the experiments were repeated at least three times, and representative experiments are shown. The protein bands were quantified using ImageJ (National Institutes of Health, Bethesda, MD, USA).

### 2.5. Quantitative Reverse Transcription PCR (RT-qPCR)

EA.hy926 cells or HUVECs were seeded in 6-well plates and cultured until they reached 80% confluence. The medium was replaced with a serum-free medium for a 4 h starvation. For EA.hy926 cells, the compounds were added at a final concentration of 0.5 μM for 30 min, followed by BMP9 (1 ng/mL), TGF-β2 (1 ng/mL), or vehicle control addition and cultured for another 3 h before the cells were harvested. To investigate VEGF-induced cellular response, HUVECs were exposed to VEGF (40 ng/mL) for 1 h after 30 min pre-treatment with the compounds.

After samples were collected, total RNA was isolated and purified using the NucleoSpin RNA II kit (740955, BIOKE, Leiden, the Netherlands), following the manufacturer’s instructions. Then, the mRNA was used as a template to synthesize cDNA by performing reverse transcription with the RevertAid First Strand cDNA Synthesis Kit (K1621, ThermoFisher Scientific). Gene expression levels were determined by RT-qPCR analysis using the GoTaq qPCR Master Mix (A6001, Promega, Madison, WI, USA) and normalized to *GAPDH* expression. All the experiments were repeated three times, and results from three biologically independent experiments are shown. Appendix A includes a list of the primer sequences used.

### 2.6. Kinase Profiling Assay

The kinase inhibition profile of the compounds was determined by testing 96 protein kinases’ activity using a radiometric protein kinase assay (33 PanQinase^®^ Activity Assay, Proqinase, Germany). In brief, 10 µL of non-radioactive ATP solution (in H_2_O), 25 µL of assay buffer/[γ-33 P]-ATP mixture, 5 µL of 0.1 μM the tested compound in 10% DMSO and 10 µL of enzyme/substrate were mixed in 96-well FlashPlatesTM (Perkin Elmer, Boston, MA, USA). The reaction was conducted at 30 °C for 60 min. To stop the reaction, 50 µL of 2% (*v/v*) H_3_PO_4_ was added into the protein kinase reaction cocktails. After which, with 200 µL 0.9% (*w/v*) NaCl twice, the incorporation of ^33^ Pi (counting of “cpm”) was determined with a microplate scintillation counter (Wallac Microbeta, Boston, MA, USA). All protein kinase assays were performed using a BeckmanCoulter Biomek 2000/SL robotic system (Brea, CA, USA). The residual activity (in %) for each compound was normalized to 100% enzyme activity (untreated control) and background (negative control).

### 2.7. Biochemical Affinity Assessment

The binding interaction profile of the compounds for ALK1–6, ACVR2A, ACVR2B, and TGFβR2 was determined by using radiometric protein kinase assay. For each compound, ten different concentrations (from 3 × 10^−6^ M to 9 × 10^−11^ M) were used in the protein kinase reaction. The inhibitor binding constants (Kd values) were calculated based on the 10 corresponding residual activities for each compound using Prism 5.04 (Graphpad, San Diego, CA, USA) according to the following formula:Kd = ([K][I]/[C])
where [K] = molar concentration of non-inhibitor bound kinase at equilibrium, [I] = molar concentration of the free inhibitor at equilibrium, [C] = molar concentration of kinase-inhibitor complex at equilibrium.

### 2.8. Migration Assay

Approximately 5 × 10^4^ EA.hy926 cells/well were seeded in a 96-well microplate Essen ImageLock™ (4379, Essen Bioscience, Ann Arbor, MI, USA) and cultured overnight. Then homogeneous 700–800 micron-wide scratch wounds were made using a WoundMaker™ device (4563, Essen Bioscience, Ann Arbor, MI, USA) containing 96 pins. The compounds and/or ligands were added to each well in 100 µL of fresh serum-free medium. Each group contained 6 replicates. Real-time pictures were acquired every 4 h by the IncuCyte ZOOM (Essen Bioscience) to track the wound closure. The wound density was analyzed using the software package provided by the manufacturer and normalized to the wound width at the start of the experiment. All the experiments were repeated at least three times, and representative results are shown.

### 2.9. Proliferation Assay

Around 4 × 10^3^ EA.hy926 cells per well were seeded in a 96-well microplate Essen ImageLock™ in 200 µL of culture medium containing the indicated concentrations of compounds and ligands, and the plate was directly inserted into the IncuCyte ZOOM instrument for scanning. Images of each well were acquired every 4 h for 3 days. The real-time cell confluence in each well was analyzed after normalizing to the initial cell density. All the experiments were repeated at least three times, and representative results are shown.

### 2.10. MTS Assay

The MTS assay was used to measure the metabolic activity and viability of the cells. Around 5 × 10^3^ EA.hy926 cells were seeded and cultured overnight in a 96-well plate and then cultured in the normal growth medium, which contained the indicated compounds and ligands. The MTS tetrazolium was measured according to the manufacturer’s protocol using an MTS Assay Kit (G3580, Promega). In brief, after exposing the cells in set conditions for 1, 2, and 3 days, 20 µL of MTS reagent was added to 100 μL cell growth medium per well and incubated for 4 h at 37 °C. Thereafter, absorbance at 490 nm was measured using a PerkinElmer luminometer. All the experiments were repeated at least three times, and representative experiments are shown.

### 2.11. Cord Formation Assay

The cord formation assay was performed according to the IncuCyte Angiogenesis PrimeKit Assay protocol (Cell Player Angiogenesis PrimeKit, Essen BioScience). In brief, GFP labeled HUVEC cells and human dermal fibroblasts were mixed in a 1:20 ratio and seeded per well in a 96-well plate (3150 cells in total). Thereafter, the cells were exposed to VEGF alone or in combination with the compounds (at indicated concentrations) in a complete assay medium. Pictures were captured every 6 h for up to 9 days To monitor the cord formation. The medium was refreshed every 2 or 3 days. Cords formation was visualized by the GFP fluorescence, and cord length was quantified with the instrument’s software. All the experiments were repeated at least three times, and representative experiments are shown.

### 2.12. Chemotactic Invasion Assay

The invasion potential of ECs was investigated using the chemotactic invasion assay, according to the IncuCyte instruction manual. Briefly, the top layer of an IncuCyte ClearView 96-well chemotaxis plate (4582, Essen Bioscience) was coated with 1% gelatin for 30 min. One thousand five hundred HUVEC cells/well were seeded on the gelatin-coated top plate and covered with 60 µL M199 medium with 1% FBS containing 0.5 µM of the indicated compounds or vehicle control (DMSO). Then 200 µL/well of M199 supplemented with 20% FBS, 100 IU/mL penicillin/streptomycin, 0.01 IU/mL Heparine, and BPE with 0.5 µM of the indicated compounds was added to the bottom plate. The plate was placed in the IncuCyte, and whole well images of both top and bottom membranes were captured every 4 h for one day using the IncuCyte Chemotaxis imaging protocol. Finally, the top and bottom cell areas were quantified with the instrument’s software using automated algorithms. The ECs invasive behavior was presented as total bottom cell area normalized to the initial top cell area. All the experiments were repeated at least three times, and representative results are shown.

### 2.13. Zebrafish Embryo Toxicity Assay

Breeding pairs of the transgenic Casper zebrafish line Tg (*fli*:EGFP) were put together in a tank, and the next morning, all embryos were collected as previously described [37]. Fertilized eggs that were at the same development stage (around 2.5 h post-fertilization (hpf)) were selected under the microscope. The eggs were randomly divided into different groups, and each group contained about 20 embryos. All the embryos were raised in egg water (60 μg/mL sea salt) and placed in a 28 °C incubator. The compounds dissolved in DMSO were diluted in egg water to a final concentration of 0.5 µM. Equal amounts of DMSO were added to the vehicle control group. Egg water was changed every day, and abnormal embryos were removed. Embryos were recorded until day 3. All experiments were performed in technical triplicates and repeated three times, and representative results are shown. The survival rate was calculated according to the following formula: Survival rate (%) = 100% × (Number of zebrafish that survived at the end)/(Number of zebrafish present at the beginning).

### 2.14. Zebrafish Dorsal Longitudinal Anastomotic Vessel (DLAV) and Intra Segmental Vessel (ISV) Formation

The Tg (*fli*:EGFP) zebrafish embryos were collected, and embryos with the same development stage (around 2.5 hpf) were selected. The eggs were randomly separated into different groups, and each group contained at least 20 embryos. The embryos were exposed to egg water containing 0.5 µM compounds or vehicle control (equal amount of DMSO) for 20 h at 28 °C. Next, all the eggs were dechorionated under the microscope using tweezers. The zebrafish embryos were fixed in 4% paraformaldehyde (PFA) for 2 h at room temperature and stored at 4 °C in phosphate-buffered saline (PBS). The entire zebrafish embryo was imaged with an Andor Dragonfly 500 spinning disk confocal (Oxford Instruments, Abingdon, UK) using a 488 nm laser. Five embryos in each group were randomly selected and scanned. The zebrafish vasculature formation of DLAV and ISV after 20 h treatment with compounds was quantified using ImageJ as previously described [38]. The experiments were repeated twice with similar results.

### 2.15. Zebrafish Subintestinal Vessel (SIV) Formation

The Tg (*fli*:EGFP) zebrafish embryos were collected, and embryos with the same development stage (around 2.5 hpf) were selected. Groups containing at least 25 embryos were treated with the indicated compounds at 0.5 µM for 3 days. The egg water was refreshed every day, and the abnormal embryos were removed. After 3 days of treatment, the zebrafish embryos were fixed in 4% PFA for 2 h at room temperature and stored at 4 °C in PBS. The area containing SIV in each zebrafish embryo was scanned with an Andor Dragonfly 500 spinning disk confocal by using a 488 nm laser. At least ten zebrafish embryos in each group were randomly selected and analyzed. The SIV vasculature of the zebrafish was quantified with ImageJ. The leading buds were counted as previously described [39]. The experiments were repeated twice with similar results.

### 2.16. Tumor Angiogenesis in Zebrafish

Two days post fertilization (dpf), four hundred mCherry-labeled human breast cancer MDA-MB-231 cells were injected into the perivitelline space of Tg (*fli*:EGFP) zebrafish embryos as previously described [40]. The properly injected zebrafishes were randomly divided into five groups, and each group contained at least 30 embryos. The zebrafish embryos were maintained in egg water containing vehicle control (DMSO) or 0.5 µM compounds (including Axitinib, LDN-193189, OD16, and OD29) at 33 °C for 3 days. The egg water was refreshed daily, and the abnormal embryos were removed. After 3 days, the embryos were fixed in 4% PFA for 2 h at room temperature and stored at 4 °C in PBS. The tumor cell-induced angiogenesis in the perivitelline space was imaged with an Andor Dragonfly 500 spinning disk confocal using 488 nm and 561 nm lasers. At least ten embryos per group were randomly selected and scanned. The experiments were repeated twice with similar results. Tumor angiogenesis was quantified using ImageJ, and the relative vasculature density was calculated according to the following formula: Relative vasculature density (%) = 100% × (Vessel intensity within the perivitelline space)/(tumor cells intensity within the perivitelline space).

### 2.17. Statistical Analyses

Results were compared by an unpaired Student’s *t*-test. Differences were considered significant when *p* < 0.05.

## 3. Results

### 3.1. Identification of Two Novel Macrocyclic Compounds as BMPRI Inhibitors

Nanocyclix^®^ (Oncodesign, Dijon, France) is a unique technology platform using a chemical biology-based approach to generate and optimize macrocyclic molecules. The process starts from the Nanocyclix^®^ diverse chemistry, which delivers high-quality probes that are potent type 1 kinase inhibitor macrocycles with initial selectivity against a small “signature subset” of kinases. Extensive profiling against the human kinome and in early absorption, distribution, metabolism, and excretion (ADME) identified probes of interest, with initial potency and selectivity against a small number of kinases. Due to their small size, macrocyclic compounds cross the cell membrane and can be used as in vitro and ex vivo tools. Via this chemical diversity approach, two macrocyclic compounds, OD16 and OD29, were generated with strong inhibitory potential against kinase activity of specific BMPRI members. Interestingly, OD29 displayed a chemical structure close to the well-characterized BMPRI receptor kinase inhibitor LDN-193189 [41] (Figure 1A–C). The liquid-chromatography mass-spectrometry (LCMS) and ^1^H-NMR characterization of OD16 and OD29 are shown in Appendix A–D. The target and off-target hits of OD16 and OD29 were determined against a panel of 96 kinases, and the kinome trees showed their potential kinase targets (Figure 1A–C and Appendix A). To further assess their affinity, we tested the activity against different TGF-β and BMP type I and type II kinase receptors (Figure 1D). Promisingly, unlike LDN-193189, which has strong inhibitory effects on ALK1–5, ACVR2A, ACVR2B, and TGFβR2, OD16 and OD29 both showed much higher affinity to ALK1 and ALK2 (Figure 1D). To gain insight into how OD16 and OD29 improved the affinity to BMPRI, especially to ALK1, we modeled the interaction of the compounds to the ATP pocket of the ALK1 kinase domain (Figure 1E,F). As compared to LDN-193189, OD16 and OD29 appear to bind deeper into the ATP pocket of ALK1 kinase (Appendix A–C).

### 3.2. OD16 and OD29 Specifically Inhibit BMP, but Not TGF-β Signaling

To further validate the inhibitory properties of OD16 and OD29, we determined and compared the effect of these two compounds and LDN-193189 side by side on TGF-β family-induced signaling responses, i.e., BMP9 and BMP6-induced SMAD1/5 phosphorylation and TGF-β2-induced SMAD2 phosphorylation, SMAD-dependent transcriptional activity, and target gene expression. To test this, we first characterized the BMPRI expression levels in the human umbilical vein cell line EA.hy926. ALK1 and ALK2 were found to be highly expressed in EA.hy926, while ALK3 and ALK6 were expressed at low levels, comparable to HUVECs (Appendix A). The EA.hy926 cells were stimulated with BMP9, a high-affinity ligand for ALK1, and is likely to signal mainly via ALK1 as previously shown for HUVECs [19,42]. As expected, BMP9 triggered the phosphorylation of SMAD1/5 after 1 h stimulation, which was inhibited in a dose-dependent manner in the presence of LDN-193189 (0–1.6 μM), without affecting the total SMAD1 levels (Figure 2A). Treating EA.hy926 cells with different doses of OD16 or OD29 revealed that BMP9-activated SMAD1/5 phosphorylation was inhibited by both compounds (Figure 2B,C). The half-maximal inhibitory concentration (IC50) values of OD16 and OD29 were 0.08 μM and 0.43 μM, respectively, while the IC50 of LDN-193189 was 0.11 μM (Figure 2D,E). Thus, OD16 is more potent than OD29 and LDN-193189 in inhibiting BMP9-induced SMAD1/5 phosphorylation, a response that is likely mediated by ALK1 in these cells.

Next, we compared the effect of LDN-193189, OD16, and OD29 on BMP/SMAD signaling using the SMAD1/5-SMAD4 dependent BRE-luciferase (Luc) transcriptional reporter assay (Figure 2F). As expected, BMP9 markedly increased the BRE-Luc activity with a 5.0 fold increase, and this effect was antagonized by LDN-193189 treatment. BMP9-induced BRE activity was significantly inhibited at 0.5 μM and nearly decreased to basal levels at 1 μM. OD16 already strongly suppressed the BMP9-induced BRE-Luc activity from 5.0 to 2.1 fold at a low dose (0.1 μM) and, similar to LDN-193189, nearly completely inhibited BRE-activity at 1µM. OD29 repressed BMP9-induced BRE-Luc activity in a dose-dependent manner. For example, 0.5 μM treatment decreased activity from 5.0 to 3.1 fold, and 1 μM blocked the BRE activity to 0.9 fold (Figure 2F). Thus, consistent with its effects on BMP9-induced SMAD1/5 phosphorylation, OD16 is more potent than LDN-193189 and OD29 in inhibiting BMP9-induced BRE-Luc activity. These results were further validated by the analysis of the effects of LDN-193189, OD16, and OD29 on the BMP9-induced upregulation of *ID1*, *ID3,* and *SMAD6* mRNA expression. All three compounds inhibited the BMP9-induced gene responses (Figure 2G–I). OD16 demonstrated a stronger inhibitory effect than LDN-193189, while LDN-193189 was significantly more potent than OD29.

Next, we stimulated the EA.hy926 cells with BMP6, which mainly signals via ALK2 and does not bind and signal via ALK1 [43], and interrogated the effects of LDN-193189, OD16, and OD29 on signaling mediated via ALK2. LDN-193189 and OD16 efficiently inhibited BMP6-induced SMAD1/5 phosphorylation at a low dose (0.1 µM) (Figure 2J,K). OD29, however, failed to antagonize BMP6 induced pSMAD1/5 at a low concentration (0.1 µM). At higher concentrations (1 µM), a very weak inhibitory effect of OD29 was observed. Quantification of the pSMAD1/5 levels normalized to total SMAD1/5 is presented in Figure 2M.

To further investigate and validate the selective inhibitory effects of OD16 and OD29 toward BMP9/ALK1 and BMP6/ALK2 signaling, we transfected C2C12 myoblast cells with constitutively active mutant ALK1 and ALK2 (caALK1 and caALK2). These constitutively active receptors signal in the absence of exogenous ligands and allow for a more direct assessment of inhibitory effects that are elicited by kinase receptor inhibitors [44]. We tested the effects of LDN-193189, OD16, and OD29 compounds on the caALK-induced BRE-Luc activity. As shown in Appendix A, the ectopic expression of caALK1 and caALK2 significantly increased BRE luciferase activity, and LDN-193189 in a dose-dependent manner attenuated both caALK1 and caAL2-induced BRE-Luc reporter activity. No difference in inhibition between ALK1 and ALK2 was observed for LDN-193189. Of note, consistent with the Western blot results, OD16 and OD29 exhibited higher sensitivity and more inhibitory potential against caALK1 compared to caALK2 (Appendix A).

To further examine the specificity of the OD16 and OD29 compounds, we examined their effects, compared to LDN-193189, on TGF-β2 activated SMAD2 phosphorylation response in ECs. As shown in Figure 2N, LDN-193189 effectively attenuated TGF-β2-induced SMAD2 phosphorylation at a low dose (0.1 µM). Treatment of cells with OD16 inhibited the TGF-β2-induced pSMAD2 in a dose-dependent manner but far less efficiently than LDN-193189 (Figure 2O). Interestingly, OD29 demonstrated nearly no inhibitory effect on TGF-β2-activated SMAD2 phosphorylation (Figure 2P,Q). Thus, the effects of LDN-193189 on TGF-β signaling are more apparent than OD16, while OD29 has no significant effect on the TGF-β-induced transcription response.

In conclusion, both OD16 and OD29 are efficient BMPRI inhibitors. OD16 efficiently inhibits ALK1 and ALK2 signaling, with higher potency than LDN-193189, while it has an inhibitory effect on TGF-β signaling. Whereas OD29 is less potent than LDN-193189 and OD16 in antagonizing BMP signaling, however, it has a more selective inhibitory effect on ALK1 than on the closely related ALK2. Importantly, OD29 had no significant inhibitory effects on TGF-β/SMAD signaling.

### 3.3. OD29 Antagonizes VEGF-Induced ERK MAP Kinase

Angiogenesis is strongly induced by vascular endothelial growth factor (VEGF) [4,42,45]. Certain BMPRI kinase inhibitors have been found to also target VEGFR kinase or downstream MEK kinase activity [46,47]. We, therefore, examined whether OD16 and OD29 affect VEGF-induced signaling responses in ECs. HUVECs are highly sensitive to VEGF stimulation, as demonstrated by the ability of VEGF to induce VEGFR and ERK phosphorylation (Figure 3A). Moreover, Axitinib, a VEGFR1/2/3 kinase inhibitor, blocked the VEGF-induced pERK response (Figure 3A) [48]. While OD16 failed to alter VEGF-induced pERK (Figure 3B), treatment of HUVECs with OD29 profoundly decreased VEGF-stimulated pERK in a dose-dependent manner (Figure 3C,D). Of note, unlike Axitinib, OD29 did not affect VEGF-induced pVEGFR. To further validate the inhibitory function of OD29 on VEGF signaling, we investigated its effect on VEGF-induced *DLL4* and *NR4 A* mRNA expression [49,50] (Figure 3E,F). As shown in Figure 2E, both Axitinib and OD29 inhibited VEGF-induced *DLL4* and *NR4 A1* mRNA expression. Surprisingly, OD16, which had no effect on VEGFR signaling, did inhibit VEGF-induced *DLL4*, suggesting a cooperative cross-talk between VEGF and BMP intracellular signaling on the expression of these target genes [51]. The inhibitory effect of OD29 on BMP receptor signaling may, therefore, also contribute to its effect on VEGF-induced target gene expression. In conclusion, besides BMP signaling, OD29 also antagonizes VEGF-induced phosphorylation of ERK MAP kinase.

### 3.4. OD16 and OD29 Antagonize ECS Migration, Cord Formation, and Invasion

Given the inhibitory effects of OD16 and OD29 on BMP or/and VEGF signaling that have pivotal roles in the regulation of EC function, we examined the effects of OD16 and OD29 on EC behavior, including cell proliferation, migration, cord formation, and invasion. The effect of OD16 and OD29 on cell proliferation was measured using a MTS assay and cell surface coverage by IncuCyte imaging. As shown in Appendix A, OD16 and OD29 exhibited no significant effects on EA.hy926 cell viability, in common with Axitinib, while LDN-193189 markedly attenuated cell viability. Analysis of the real-time images of EA.hy926 ECs revealed that LDN-193189 strongly inhibited ECs proliferation, unlike OD16 and OD29 (Appendix A). Subsequently, scratch wound healing assays were performed to assess the effect of OD16, OD29, LDN-193189, and Axitinib on the EC migration. As shown in Figure 4A, the real-time relative wound density in each group was quantified, and LDN-193189, OD16, or OD29 significantly inhibited the BMP9-induced EA.hy926 ECs migration compared to vehicle control (DMSO) after 24 h. Representative images are presented in Appendix A. In addition, we assessed the influence of the compounds on VEGF-induced EC migration. As shown in Figure 4B and Appendix A, whereas LDN-193189 had no inhibitory effect on VEGF-induced cell migration, OD29 and OD16 both antagonized VEGF-induced cell migration. We also tested the effects of the compounds on fetal bovine serum (FBS)-induced EC migration (Figure 4C and Appendix AC). While LDN-193189 and OD16 did not affect FBS-stimulated EC migration, OD29 was effective in suppressing FBS-induced ECs migration. These results suggest that both OD16 and OD29 inhibit EC migration, likely through different underlying mechanisms.

To further evaluate the effect of OD16 and OD29 on angiogenesis, the effects of the compounds on EC cord formation were investigated. As shown in Figure 4D, when co-cultured with fibroblasts, the eGFP-labeled HUVECs formed capillary-like cord structures. Upon challenging them with VEGF, more EC cord structures were formed. As expected, the VEGF inhibitor Sunitinib strongly suppressed VEGF-induced EC cord formation [52]. Consistent with a previous report, LDN-193189 showed no inhibitory effect and even slightly enhanced VEGF-mediated cord formation [53]. In contrast, the two macrocyclic compounds OD16 and OD29 antagonized VEGF-derived cord formation, with OD29 being slightly more potent than OD16 (Figure 4D). The quantified network area and network length of all the groups are shown in Appendix A. To further elucidate the inhibitory effect of OD16 and OD29 on EC cord formation, we exposed the co-cultured cells to different concentrations of the compounds. As shown in Figure 4E, VEGF augmented 6.8 times the total length of the formed tubes compared to the group without VEGF treatment. The VEGF inhibitor Axitinib potently inhibited this VEGF-induced response in a dose-dependent manner (Appendix A). The real-time changes of different concentrations of Axitinib on VEGF-induced tube length are shown in Appendix A. OD16 and OD29 antagonized in a dose-dependent manner the VEGF-induced cord formation (Figure 4E). The tracking of the tube length changes of OD16 and OD29 presented groups are shown in Appendix A. OD16 displayed apparent inhibitory effects at 0.5 μM, while OD29 already effectively attenuated VEGF-induced EC cord formation at 0.1 μM. These results demonstrated both OD16 and OD29 share robust inhibitory potential on VEGF-stimulated EC cord formation. Next, we examined the influence of OD16 and OD29 on ECs’ invasion ability. Both Axitinib and LDN-193189 significantly compromised HUVECs movement as fewer cells appeared at the bottom of the membrane (Figure 4F). Both OD16 and OD29 similarly blocked the invasion of HUVECs. Taken together, the two macrocyclic compounds OD16 and OD29 show a promising potential to inhibit angiogenesis.

### 3.5. The Macrocyclic OD16 and OD29 Inhibit DLAV, ISV, and SIV Formation in Zebrafish

In order to explore the effects of the macrocyclic compounds OD16 and OD29 on zebrafish vascularization, we treated fertilized zebrafish embryos with the compounds for 20 h and examined their DLAV and ISV formation (Figure 5A). The VEGFR inhibitor Axitinib and the BMPRI inhibitor LDN-193189 were used as comparison. Embryos around 2.5 hpf were used (Figure 5B). First, we tested for potential toxic effects of the compounds on the zebrafish embryos. Upon exposure of the larvae to the egg water containing OD16 and OD29 for three days, the survival rate of the zebrafish was around 80–93% and similar to the control group (Figure 5C). After treatment of the embryos at the same developmental stage with the inhibitors for 20 h, the effects on the formation of DLAV and ISV were assessed (Figure 5D). While in the control group, the SIV sprouted from the dorsal aorta and connected with a well-formed DLAV (Figure 5D, upper panel), Axitinib-treated zebrafish, as previously reported [54], exhibited strong defects in both DLAV and ISV formation. The zebrafish embryos exposed to LDN-193189 did not exhibit apparent defects in DLAV and ISV development. Interestingly, OD16 strongly delayed the DLAV formation process and slightly inhibited ISV development compared with the control group. Similarly, OD29 had a strong negative impact on vascular network formation, especially with respect to DLAV development. The quantification of the DLAV and ISV vascularization is depicted in Figure 5E. The number of junctions indicated the connectivity of ISV (Figure 5F). Axitinib strongly inhibited the total vessel length and connectivity, while LDN-193189 showed no effects. OD16 and OD29 significantly inhibited the DLAV and ISV formation and ISV junctions, although the inhibition effect was not as strong as observed for the VEGF inhibitor Axitinib. In contrast, the BMPRI inhibitor LDN-193189 did not influence the early vessel development in zebrafish.

To further investigate the anti-angiogenic potential of OD16 and OD29, we assessed their influence on SIV formation at a relatively late stage of zebrafish embryonic development (Figure 6A). After exposure of the 2.5 hpf embryos to the compounds for 3 days, the SIV formation was examined. In vehicle control-treated zebrafish embryos, intact and well-shaped SIV developed after 3 dpf (Figure 6B upper panel). In the presence of Axitinib, a strong reduction in the number of SIVs and serious other vascular defects were observed, consistent with previous reports [54]. In the LDN-193189 treatment group, the SIV was slightly less well-developed than in the control group. Of note, LDN-193189 induced an abnormal phenotype of the vessels, as numerous ectopic SIV tip cells were observed (Figure 6B labeled with stars). In contrast, a significant decrease in the number of ectopic vessels and SIV tip cells was observed in OD16 treated embryos. Similarly, the embryos treated with OD29 exhibited an anti-angiogenic SIV phenotype. We quantified the SIV vessel length and the number of tip cells in each group (Figure 6C,D). As shown in Figure 6C, both OD16 and OD29 significantly inhibited vessel formation as compared to vehicle control-treated zebrafish embryos. Axitinib markedly inhibited angiogenesis, and LDN-193189 treatment resulted in a lack of SIV vessels compared with the control group. Interestingly, unlike Axitinib, OD16 and OD29 did not influence the formation of tips. LDN-193189 significantly increased the number of disorganized SIV tip cells, which might drive the subsequent ventral expansion of the SIV (Figure 6D). These results support the idea that OD16 and OD29 inhibit angiogenesis in vivo.

### 3.6. OD16 and OD29 Inhibit Breast Cancer-Induced Vessel Invasion in Zebrafish

Since OD16 and OD29 function as BMPRI kinase inhibitors and OD29 also inhibits VEGF receptor signaling in ECs, we sought to determine whether these compounds could inhibit tumor angiogenesis. To test this, we injected mCherry-labeled human MDA-MB-231 breast cancer cells into the perivitelline space of 2 dpf Tg (*fli*:EGFP) zebrafish embryos, in which blood vessels are labeled fluorescent green. In a 3-day period, the tumor cells induced the formation and growth of blood vessels within the perivitelline space of the embryos (Figure 7A). Whereas Axitinib markedly inhibited the cancer cell-induced angiogenesis, LDN-193189 displayed a minor although significant effect (Figure 7B). Interestingly, both OD16 and 0 D29 significantly inhibited the formation of vessels surrounding the injected breast cancer cells (Figure 7C). In summary, our results demonstrate that OD16 and OD29 are two novel macrocyclic BMPRI kinase inhibitors with anti-tumor angiogenesis activity. Following our previously reported macrocyclic compounds with enhanced activity against ALK2 [33], we demonstrated here the potential of macrocyclization as a new chemical approach to obtain BMPR-specific inhibitors with promising therapeutic potential in human disease.

## 4. Discussion

TGF-β family members exert pleiotropic cellular functions, and disturbances in their signaling are often associated with disease [55]. For TGF-β ligands to induce intracellular responses, the formation and activation of specific membrane receptor complexes with intrinsic serine/threonine kinase activity are required. For example, BMP ligands signal via ALK1, ALK2, ALK3, and ALK6. Selective inhibitors of their kinase activity will be of high value to investigate their unique or redundant function in biological processes and enable therapeutic approaches. However, due to the high homology in the ATP-binding pockets in kinase domains, the development of selective BMPRI kinase ATP-competitive inhibitors has remained difficult to achieve. Here, we report two novel macrocyclic BMPRI kinase inhibitors named OD16 and OD29 that efficiently antagonize BMPRI function, including SMAD1/5 phosphorylation and downstream gene expression. In comparison to LDN-193189, the macrocyclic compounds OD16 and OD29 display an improved selectivity among BMPRIs and reduced inhibitory effects on TGF-β and other kinases [33]. OD16 and OD29 behave as selective ALK1 and ALK2 inhibitors. Moreover, both compounds demonstrated much lower activity on TGF-β signaling. Interestingly, OD29 also exhibited inhibitory effects on VEGF signaling. Thus, OD29, but not OD16, acts as a dual inhibitor of the BMP and VEGF pathways. Importantly, both OD16 and OD29 showed inhibitory effects on EC function in vitro and possessed potent anti-angiogenesis activity in vivo.

BMP and VEGF pathways are well-known regulators of endothelial cell behavior and small molecules specifically regulating these pathways have a potential value in therapeutic settings [21,56]. A wealth of evidence indicates that VEGF effectively promotes cell proliferation, migration, and tube formation in different endothelial cell types [57,58]. However, the effects of BMP9 on ECs’ growth and migration appear highly context-dependent and require further investigation [14,15]. Both the selective BMPRI inhibitor OD16 and our new dual BMP and VEGF signaling inhibitor OD29 strongly inhibited ECs’ migration induced by VEGF/BMP9 and also inhibited VEGF mediated cord formation and invasion in HUVECs (Figure 4). Our results suggest the potential of OD16 and OD29 as anti-angiogenesis agents.

We validated the effects of OD16 and OD29 in vivo using zebrafish embryos. Consistent with the results in vitro, OD16 and OD29 inhibited the development of blood vessels, as measured by changes in the DLAV, ISV, and SIV at early and late developmental stages. Interestingly, since the ALK1/2 inhibitor, OD16, inhibited vessel formation, our results indicated that the ALK1/2 pathway could stimulate angiogenesis during zebrafish embryo development. Using the dual BMP and VEGF inhibitor dorsomorphin, Hao et al. showed that the inhibition of VEGF is mainly attributed to the inhibitory effects on angiogenesis. Whereas dorsomorphin blocked ISV formation, the BMPRI inhibitor DMHI1 did not affect ISV development in zebrafish [47]. In agreement with our results, Cannon et al. showed that the BMP antagonist LDN-193189 has a minor effect on ISV formation in zebrafish and further indicated the dispensable role of BMP pathway in early vessel development in zebrafish [41]. In contrast, OD16 and OD29 inhibited the HUVECs cord formation assay. The possible reason for the different effects of these BMPRI inhibitors on angiogenesis is that LDN-193189 when compared to OD16, may have off-target effects. Our results suggest an important role of both BMP and VEGF in angiogenesis during embryonic development.

Due to the structural and functional similarity of different BMPRIs, the small molecular inhibitors (such as LDN-193189) targeting BMPRIs do not commonly show high selectivity. Our results show that OD16 and OD29 selectively antagonize ALK1 and ALK2-mediated signaling compared to other BMPRIs. Previous reports unveiled the opposite function of ALK1 in regulating ECs sprouting and angiogenesis [23,24]. Similarly, the effects of ALK2 on ECs and angiogenesis are not clearly understood yet. For example, Lee et al. showed that the endothelial-specific knockout of ALK2 blocked retinal angiogenesis in mice, indicating that ALK2 mediates the angiogenesis process [59]. However, in a different report, knockdown of ALK2 using siRNA enhanced the sprouting of HUVEC spheroids, suggesting a negative role for ALK2 in angiogenesis [4]. OD16 and OD29 efficiently antagonized ECs migration and cord formation and also inhibited vessel formation in zebrafish, suggesting that ALK1 and/or ALK2 are indispensable for regulating EC function and angiogenesis. 

The inhibitory activity of OD16 and OD29 on EC function and angiogenesis in zebrafish make them of interest for the development of therapeutic approaches of vessel-related disorders. As OD16 is a BMPRI inhibitor and OD29 is a dual BMP and VEGF pathway inhibitor, their antagonistic activity on tumor-induced angiogenesis was assessed in a breast cancer xenograft model in zebrafish. The significant inhibition of newly formed vessels surrounding breast cancer cells in zebrafish demonstrated their ability to antagonize tumor angiogenesis in vivo. It will be interesting to explore the potential of OD16 or OD29 (and analogs), alone or in combination with chemotherapy drugs, radiation, or immune therapy for the treatment of solid tumor.

## 5. Conclusions

In conclusion, we synthesized two novel macrocyclic kinase inhibitors of which OD16 targets BMPRI and OD29 inhibits both BMP and VEGF signaling. These two inhibitors antagonize ECs function and inhibit angiogenesis in both normal and tumor processes. These two macrocyclic compounds may open new avenues for developing new anti-angiogenesis cancer therapies.

## Figures and Tables

**Figure 1 cancers-13-02951-f001:**
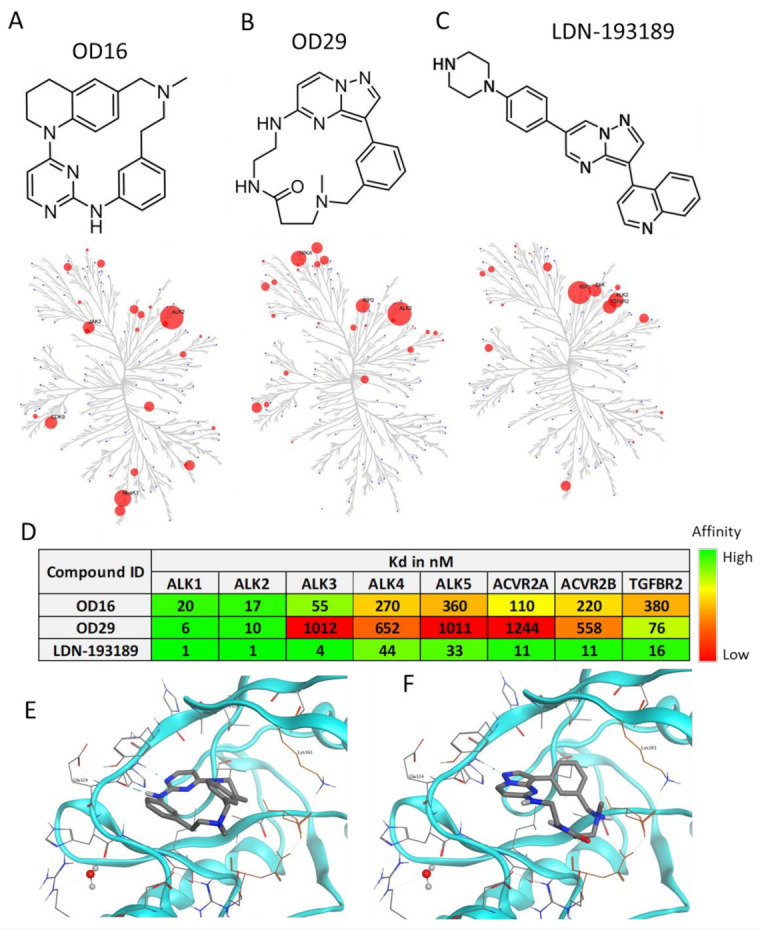
Structure and kinase profile of OD16 and OD29. (**A**–**C**) Chemical structure and the activity of a panel of 96 kinases in the presence of 100 nM of OD16 (**A**), OD29 (**B**), and LDN-193189 (**C**). (**D**) Affinity (Kd) of OD16, OD29, and LDN-193189 for the BMP, activin, and TGF-β type I and type II kinase receptors. (**E**,**F**) Structural model of OD16 (**E**) and OD29 (**F**) binding to the ALK1 hinge region in the ATP pocket.

**Figure 2 cancers-13-02951-f002:**
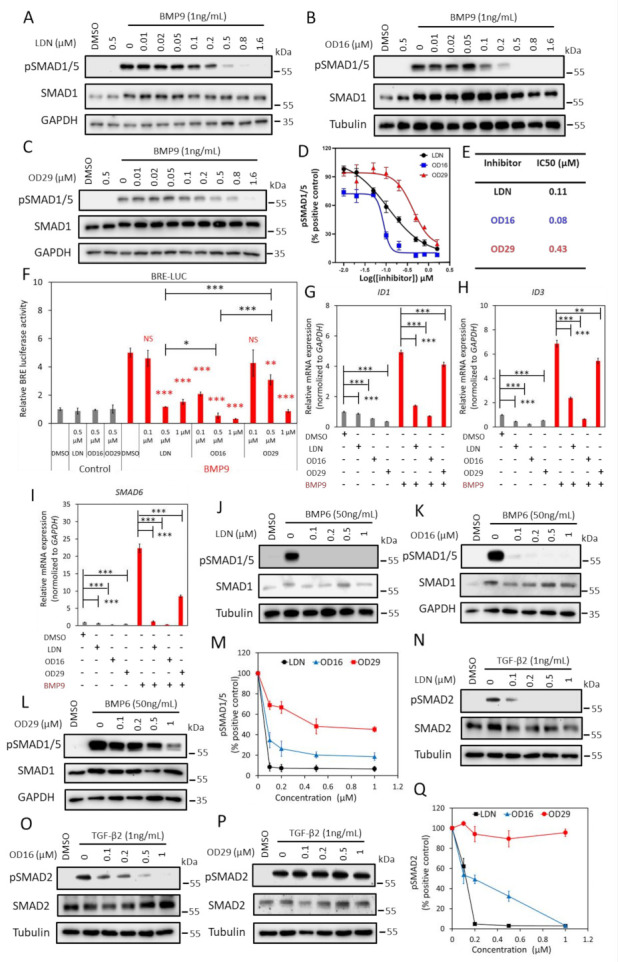
OD16 and OD29 inhibit BMP, but not TGF-β signaling in EA.hy926 cells. (**A**–**C**) Western blot analysis of the dose-dependent effects of LDN-193189 (**A**), OD16 (**B**), and OD29 (**C**) on BMP9 (1 ng/mL)-induced SMAD1/5 phosphorylation in EA.hy926 cells. (**D**) Quantification of the BMP9-induced pSMAD1/5 phosphorylation. GAPDH or Tubulin was used for protein loading controls. Results from three biologically independent experiments were integrated. The results are expressed as mean ± SD. (**E**) IC50 values of LDN-193189, OD16, and OD29 on BMP9-induced pSMAD1/5 response in EA.hy926 cells. (**F**) Inhibitory effects of LDN-193189, OD16, and OD29 on BMP9-induced BRE transcriptional reporter activity in EA.hy926 cells. Cells were preincubated with the inhibitors at three different concentrations (0.1 µM, 0.5 µM, and 1 µM) for 1 h and then stimulated with BMP9 at 1 ng/mL for 16 h with the presence of the compounds. Representative results from three biologically independent experiments are shown as mean ± SD. * *p* < 0.05, ** *p* < 0.005, *** *p* < 0.001. NS, non-significant. (**G**–**I**) RT-qPCR analysis of the effects of LDN-193189, OD16, OD29 (0.5 µM) or vehicle control (DMSO) on BMP9 (1 ng/mL)-induced *ID1* (**G**), *ID3* (**H**), and *SMAD6* (**I**) mRNA expression in EA.hy926 cells. Expression levels were normalized to those of the housekeeping gene *GAPDH*. Results from three biologically independent experiments are shown as mean ± SD. ** *p* < 0.005, *** *p* < 0.001. (**J**–**L**) Western blot analysis of the dose-dependent effects of LDN-193189 (**J**), OD16 (**K**), OD29 (**L**), or vehicle control (DMSO) on BMP6 (50 ng/mL)-induced SMAD1/5 phosphorylation in EA.hy926 cells. (**M**) Quantification of the BMP6-induced SMAD1/5 phosphorylation. GAPDH or Tubulin was used as loading controls. (**N**–**P**) Western blot analysis of the dose-dependent effects of LDN-193189 (**N**), OD16 (**O**), OD29 (***p***), or vehicle control (DMSO) on TGF-β2 (1 ng/mL)-induced SMAD2 phosphorylation in EA.hy926 cells. (**Q**) Quantification of the TGF-β2-induced SMAD2 phosphorylation. Tubulin was used as a loading control. The results from three biologically independent Western blot experiments were integrated. The results are expressed as mean ± SD. Complete Western Blot images of subfigures (**A**–**C**) and (**J**–**L**,**N**–**P**) are available in Appendix A.

**Figure 3 cancers-13-02951-f003:**
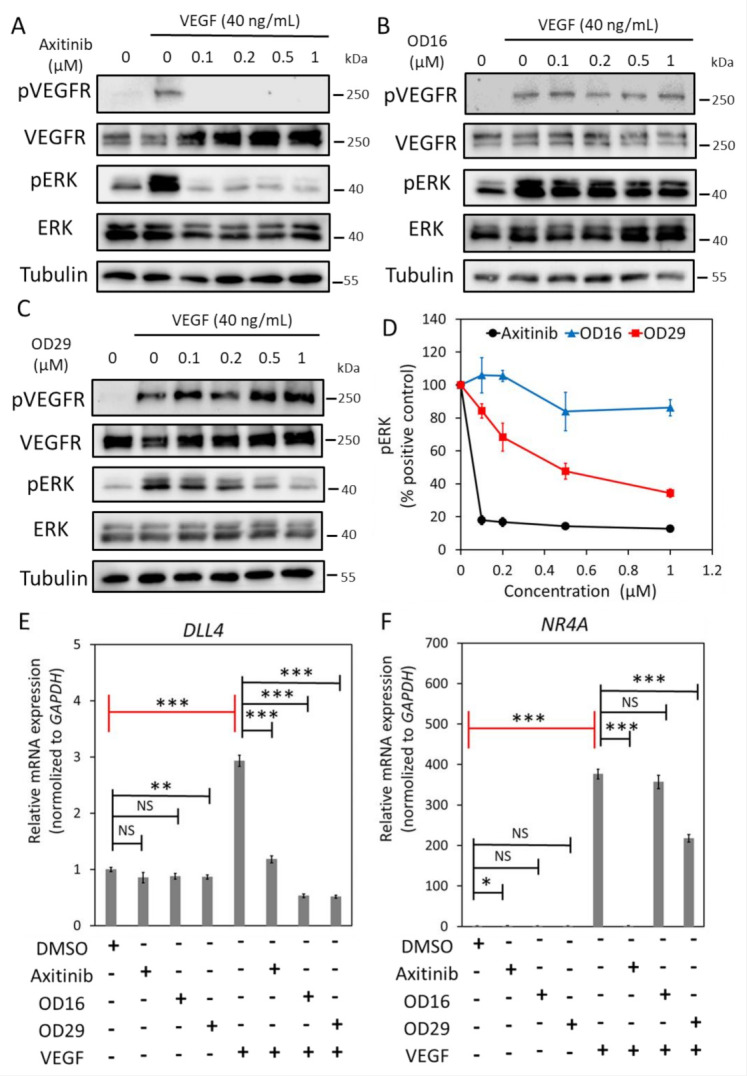
OD29 antagonizes VEGF-induced pERK MAP kinase in HUVECs. (**A**–**C**) Western blot analysis of dose-dependent effects of small molecule VEGFR kinase inhibitor Axitinib (**A**), OD16 (**B**), OD29 (**C**), or vehicle control (DMSO) on VEGF (40 ng/mL)-induced VEGFR and ERK MAP kinase phosphorylation in HUVECs. (**D**) Quantification of the Western blot analysis of VEGF-induced p-ERK MAP kinase. Tubulin was used as a loading control. The results from three biologically independent Western blot experiments were integrated. The results are expressed as mean ± SD. (**E**,**F**) RT-qPCR analysis of the effects of Axitinib, OD16, OD29 (0.5 µM), or vehicle control (DMSO) on VEGF (40 ng/mL)-induced *DLL4* (**E**) and *NR4 A* (**F**) mRNA expression in HUVECs. Results from three biologically independent experiments are shown as mean ± SD. * *p* < 0.05, ** *p* < 0.005, *** *p* < 0.001. NS, not significant. Complete Western Blot images of subfigures (**A**–**C**) are available in Appendix A.

**Figure 4 cancers-13-02951-f004:**
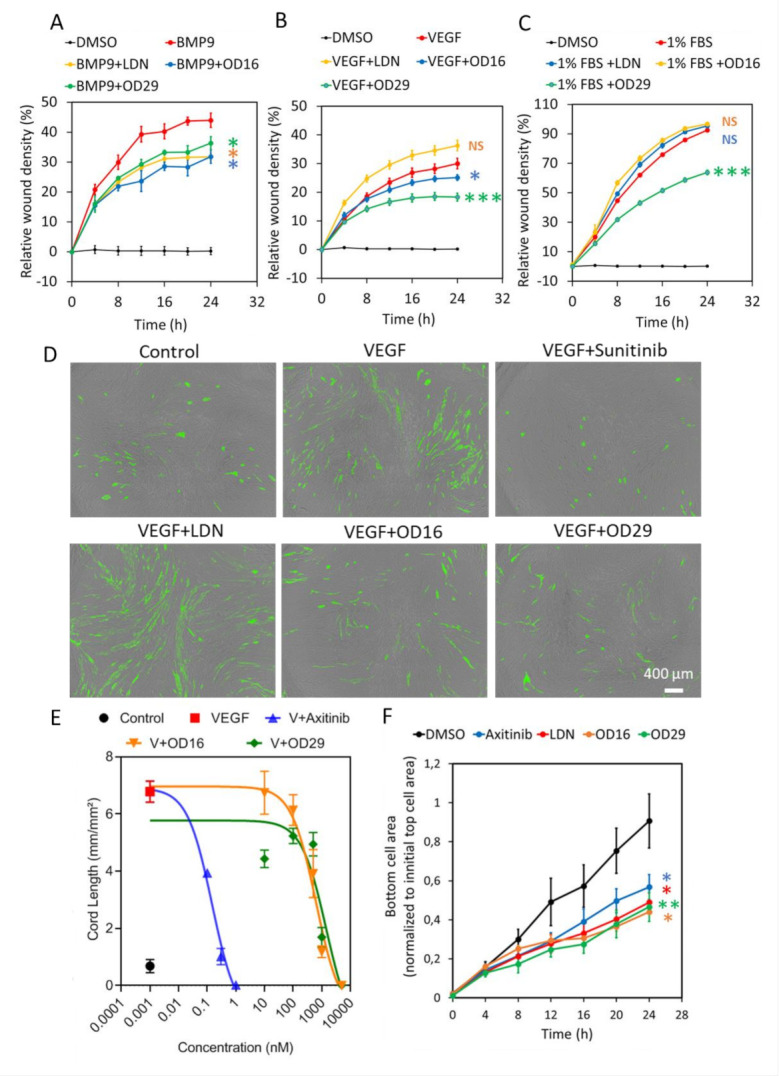
OD16 and OD29 attenuate EC migration, cord formation, and invasion in vitro. (**A**–**C**) Analysis of the real-time migration behavior of EA.hy926 cells exposed to (**A**) BMP9 (1 ng/mL), (**B**) VEGF (15 ng/mL), and (**C**) 1% FBS in the presence of vehicle control (DMSO), LDN-193189, OD16, or OD29 (0.5 µM) within 24 h. Representative results from three biologically independent experiments are shown as mean ± SD. (**D**) Cord formation assay of VEGF (15 ng/mL)-stimulated HUVEC-eGFP cells co-cultured with human dermal fibroblasts for 6 days to assess the effects of vehicle control (DMSO), OD16, or OD29 (0.5 µM). The small molecule VEGFR kinase inhibitor Sunitinib (1 μM) and BMPR-I kinase inhibitor LDN-193189 (0.5 µM) were used for comparison. Representative images are shown. Scale bar represents 400 μm. (**E**) Quantification the effects of OD16 or OD29 (0.5 µM) on VEGF (20 ng/mL)-induced cord formation HUVEC-eGFP cells co-cultured with human dermal fibroblasts over 9 days. The VEGFR kinase inhibitor Axitinib was included for comparison. The cord length is shown as mean ± SD. (**F**) Effects of vehicle control (DMSO), Axitinib, LDN-193189, OD16, or OD29 (0.5 µM) on HUVEC chemotactic cell invasion (towards full culture medium with 20% FBS and supplements). NS, not significant; * *p* < 0.05; ** *p* < 0.005, *** *p* < 0.001.

**Figure 5 cancers-13-02951-f005:**
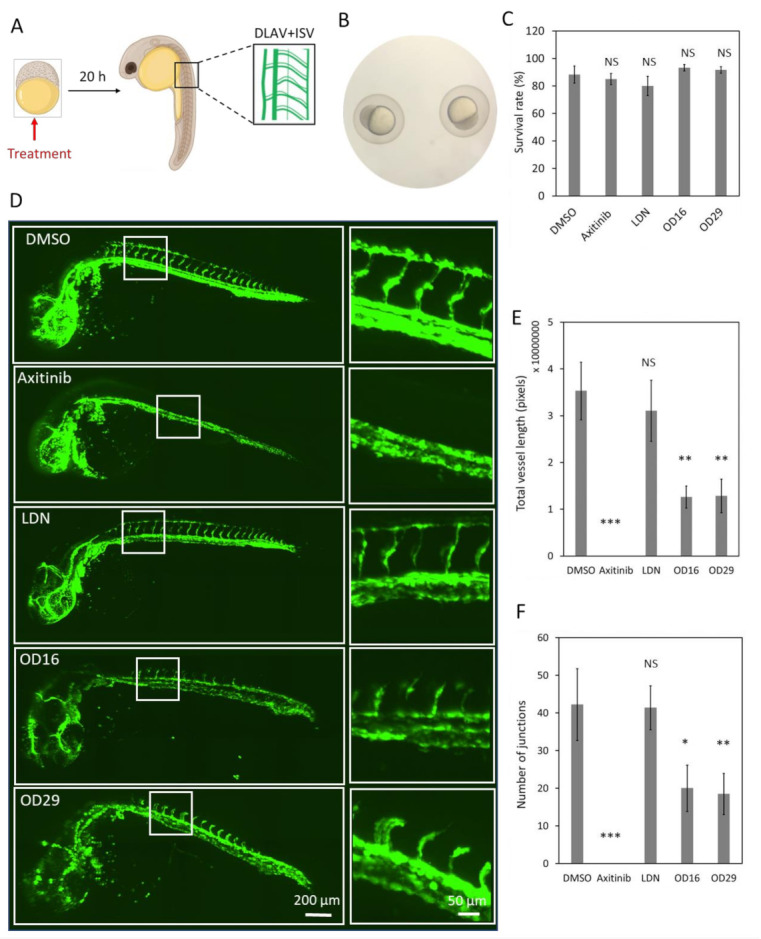
OD16 and OD29 partially inhibit DLAV and ISV formation in zebrafish embryos. (**A**) Schematic representation of the experimental procedure. The compounds were administered to the 2.5 hpf Tg (*fli*:EGFP) zebrafish eggs, and DLAV and ISV vascularization was analyzed after 20 h treatment. (**B**) Representative image of the selected zebrafish embryos that were at the same development stage (around 2.5 hpf). (**C**) Toxicity assessment of the compounds in zebrafish embryos. The zebrafish survival rate was quantified after the addition of vehicle control (DMSO), Axitinib, LDN-193189, OD16, or OD29 (0.5 µM) in egg water for 3 days. Experiments were performed three times, and representative results are shown. NS, not significant. (**D**) Effects of OD16, OD29, Axitinib, and LDN-193189 at 0.5 µM, or vehicle control (DMSO) on zebrafish DLAV and ISV development. Compounds were added to the egg water for 20 h. Representative images of overview (left panel) and high magnification (white rectangle areas from left) are shown. Scale bars represent 200 and 50 µm (left and right panel, respectively). (**E**,**F**) Quantification of the DLAV and ISV formation of each treatment group of experiments shown in (**D**); the results are represented as total vessel length (**E**) and the number of junctions (**F**), and are expressed as mean ± SD. * *p* < 0.05, ** *p* < 0.005, *** *p* < 0.001. NS, not significant.

**Figure 6 cancers-13-02951-f006:**
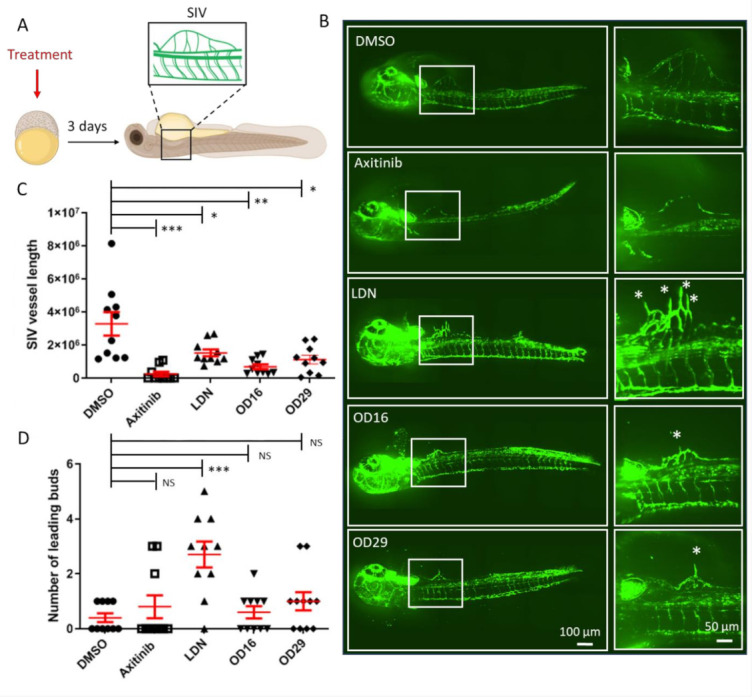
OD16 and OD29 inhibit subintestinal vessel (SIV) formation in transgenic Casper zebrafish line Tg (*fli*:EGFP). (**A**) Schematic representation of the experimental procedure. Freshly collected 2.5 hpf embryos were challenged with testing compounds in egg water. The SIV vascularization was analyzed after 3 days of treatment. (**B**) Effects of OD16, OD29, Axitinib, and LDN-193189 at 0.5 µM, or vehicle control (DMSO) on zebrafish SIV development. Compounds were added to egg water for 3 days before observing the SIV vascularization. Representative images of overview (left panel) and high magnification (white rectangle areas from left) are shown. Asterisks represent sprouting vessels. Scale bars represent 100 and 50 µm. (**C**) Quantification of the SIV vessel length of the zebrafish embryos in each group from the experiment shown in (**B**). * *p* < 0.05, ** *p* < 0.005, *** *p* < 0.001. (**D**) Quantification of the number of leading buds (indicated as asterisks in panel B). *** *p* < 0.001. NS, not significant.

**Figure 7 cancers-13-02951-f007:**
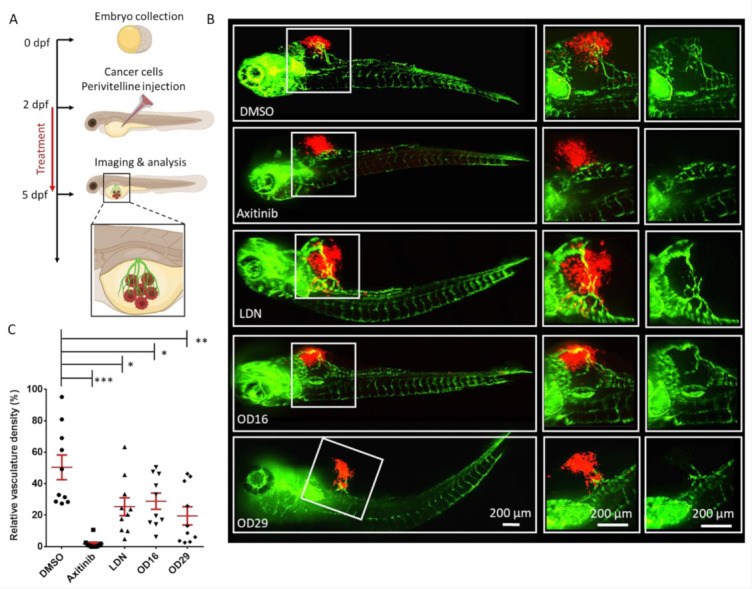
OD16 and OD29 inhibit vessel invasion in a breast cancer xenograft model in zebrafish. (**A**) Schematic representation of the experimental procedure. Two dpf zebrafish embryos were injected with mCherry (red) labeled human MDA-MB-231 breast cancer cells in the perivitelline space. The injected zebrafish embryos were challenged with testing compounds in egg water for 3 days, and the newly formed vessels within the perivitelline were analyzed. (**B**) Representative images of the eGFP expressing blood vessels (green) and mCherry MDA-MB-231 cells (red) after treatment of the zebrafish embryos with vehicle control (DMSO), Axitinib, LDN-193189, OD16, or OD29 (0.5 µM) for 3 days. The medium (vessels and tumor cells are visualized) and right panels (only vessels are visualized) represent high magnification views of the area, indicated by white rectangles in the overviews (left panel) images. Scale bars represent 200 µm. (**C**) Quantification of the relative vasculature density induced by the injected MDA-MB-231 cells within the perivitelline space of the zebrafishes in each group. * *p* < 0.05, ** *p* < 0.005, *** *p* < 0.001.

## Data Availability

The data presented in this study are available within the main manuscript and in the Appendix A.

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
