# Peer review of "Inhibiting Endothelial Cell Function in Normal and Tumor Angiogenesis Using BMP Type I Receptor Macrocyclic Kinase Inhibitors"

_cancers, 2021, doi:10.3390/cancers13122951_

Round 1
Reviewer 1 Report
In their manuscript "Inhibiting endothelial cell function in normal and tumor angiogenesis using BMP type I receptor macrocyclic kinase inhibitors" Ma and coworkers describe and characterize two novel inhibitors in the context of angiogenesis.
The methods used are adequate, and the authors present convincing in vitro and in vivo evidence for the anti-angiogenic potential of their compounds. However, since the specificity of the presented kinase inhibitors is of central importance for the conclusions, I would like to see the result of the kinase profile for the 96 kinases as described in the Methods section e.g. as table in the Supplements. A further point is that the authors claim superior affinity of their compounds towards ALKs vs. TGFBR2 in comparison to the established inhibitor LDN-193189. However, this higher affinity is not paralleled by higher specificity: the distance between ALK1 and TGFBR2 for example is approx. factor 20x for OD16 and LDN alike. The wording should be more cautious here.
Minor point: the authors should explain the calculation of their KD values.
Author Response
We thank the reviewer for her/his valuable comments. We uploaded the results of 96 kinases profile as supplementary information named “Supplementary material 2”.
To precisely describe the affinity assessment results of the compounds to the receptors (Figure 1D) as the reviewer mentioned, we rephrased “To further assess their affinity and specificity” to “To further assess their affinity” at line 347 in results. Moreover, we rephrased “To gain insight in how OD16 and OD29 improved the specificity to BMPRI” to “To gain insight in how OD16 and OD29 improved the affinity to BMPRI” at line 351 in results.
To explain how we did the calculation of the Kd value, we added more detail in the methods 2.7 biochemical affinity assessment as follows at line 209:
“The binding interaction profile of the compounds for ALK1-6, ACVR2A, ACVR2B and TGFβR2 was determined by using radiometric protein kinase assay. For each compound, 10 different concentrations (from 3 x 10-6 M to 9 x 10-11 M) were used in the protein kinase reaction. The inhibitor binding constants (Kd values) were calculated based on the 10 corresponding residual activities for each compound using Prism 5.04 (Graphpad, San Diego, California, USA) according to the following formula: Kd = ([K][I]/[C])
[K] = molar concentration of non-inhibitor bound kinase at equilibrium
[I] = molar concentration of the free inhibitor at equilibrium
[C] = molar concentration of kinase-inhibitor complex at equilibrium”

Reviewer 2 Report
The authors reported a very interesting data for anti-angiogenesis conditions. The results are very consistent and is a very nice option to initiate trials for cancer therapy based on reduction of blood vessel supply to the neoplastic mass.
Author Response
We thank the reviewer for her/his valuable comments. The manuscript was proofread by two our colleagues of which one is a native speaker in English. This improved the clarity of our manuscript.
Reviewer 3 Report
A very interesting and scientifically sound paper from a reference labororatory. My only suggestion, considering that the readers are mostly interested in oncology, is to better describe how (and in which neoplastic diseases) these BMP inhibitors can be clinically investigated.
Author Response
We thank the reviewer for her/his valuable comments. We describe the way for further possible investigation of OD16 and 0D29 in oncology in clinical as “It will be interesting to explore the potential of OD16 or OD29 (and analogues), alone or in combination with chemotherapy drugs, radiation or immune therapy for treatment of solid tumors” at the end of discussion.
The manuscript was proofread by two our colleagues of which one is a native speaker in English. This improved the clarity of our manuscript.
Round 2
Reviewer 1 Report
After the changes in the manuscript the paper can be accepted as is.